# MedAttention: A Self-Attentive RNN to Predict Diabetes Complications with Financial Data

**Rafael T. Sousa, Lucas A. Pereira, Arlindo R. Galvão Filho & Anderson da S. Soares**
Instituto de Informática
Universidade Federal de Goiás
Goiânia, Brazil
`rafaelts777@gmail.com`

## Abstract

This work is aimed to introduce MedAttention, a self-attentive recurrent neural network capable of predicting diabetes complications over health plan operators financial data. We only use financial records due to the unavailability of electronic medical records and exam results in several Brazilian health care institutions. The financial records used encodes medical procedures, hospital rates, materials and medicines. Our results succeed to predict complications over 60 to 360 days after the records processing with an area under ROC curve of 0.81 to 0.76 over time gaps. Also we introduce the possibility to visualize the attention mechanism of MedAttention to understand the patterns found.

## 1 Introduction

The last World Health Organization (WHO) diabetes report(WHO et al., 2016) point that the number of people with diabetes raised from 108 million in 1980 to 422 million in 2014, resulting on a global prevalence of 8.5% among adults over 18 years of age. The estimated number of death directly caused by diabetes in 2015 was 1.6 million(Mathers & Loncar, 2006). At the same report the WHO recognize that it is possible to prevent diabetes progress, but better mechanisms are required to identify and assess high-risk groups(WHO et al., 2016). According to the International Diabetes Federation, Brazil ranks fourth among countries with the highest number of diabetics, about 12.4 million in 2017 (Bertoldi et al., 2013; Costa et al., 2017; IDF et al., 2017).

A recent work presented a similar complication predicting technique using classic machine learning algorithms (Dagliati et al., 2017), but it relies on a complete electronic report about patients, like demographic data, body mass index, habits and results of laboratory tests. Based on the Brazilian scenario, in which electronic medical records and exam results is unavailability in several health care institutions, we propose the use of financial records, which are, among the available data, more reliable and easy to gather.

Furthermore, in the last years recent advances on Recurrent Neural Networks (RNNs) and its applications bring attention to Electronic Health Records (EHR) analysis. Recent works showed how the temporal process of RNNs can be applied to forecast medical events like diseases, hospital visits, and chronic patient complications. (Choi et al., 2016; Pandey et al., 2017; Jin et al., 2018).

Inspired by these recent works we propose a recurrent neural network architecture, based on self-attention mechanism, to predict diabetes complications through financial records of health plan operators. We call it MedAttention. Also we report an insight on how the attention mechanism can be used to highlight relevant records.

We used a dataset of financial records from the health plan operator Unimed Vitória. The dataset contains data from 2011 to 2016, from clients of Vitória's city region, which is the capital of Espírito Santo state. There is around $300,000$ individuals with 56 million records, where 2,607 was confirmed and monitored as diabetics.

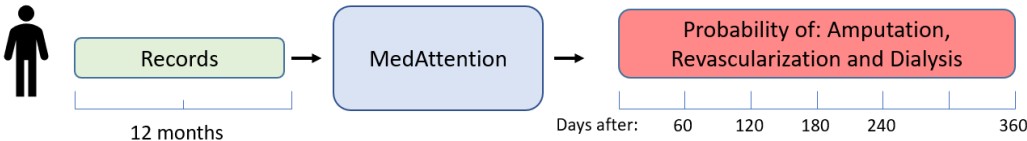

Figure 1: MedAttention architecture

Table 1: Mean results over cross-validation.

| Time Gap | MedAttention | | | LSTM | | | MLP | | |
|---|---|---|---|---|---|---|---|---|---|
| | **AUC** | Sen | Spc | **AUC** | Sen | Spc | **AUC** | Sen | Spc |
| 60 | 0,81 | 0,80 | 0,70 | 0,60 | 0,47 | 0,73 | 0,72 | 0,71 | 0,64 |
| 120 | 0,80 | 0,82 | 0,66 | 0,55 | 0,35 | 0,78 | 0,74 | 0,50 | 0,76 |
| 180 | 0,78 | 0,74 | 0,72 | 0,58 | 0,45 | 0,69 | 0,65 | 0,57 | 0,74 |
| 240 | 0,77 | 0,76 | 0,66 | 0,55 | 0,69 | 0,40 | 0,68 | 0,62 | 0,64 |
| 360 | 0,76 | 0,83 | 0,60 | 0,57 | 0,66 | 0,52 | 0,65 | 0,57 | 0,70 |

The records follow a national standard from Brazilian National Regulatory Agency for Private Health Insurance and Plans (ANS), which is called TUSS (*Terminologia Unificada em Saúde Suplementar* - Supplementary Healthcare Unified Terminology). The TUSS terminology have unique codes for: medical procedures, hospital and clinics rates, materials, medicines and special materials like orthoses and prostheses. Our target was records from three main kinds of diabetes complications: (i) Amputations and debridements; (ii) Revascularization and (iii) Dialysis. These records indicates some of the main types of diabetes complications, which is kidney failure and cardiovascular diseases.

## 2 MEDATTENTION

The proposed model is mainly inspired by the self-attention embedding proposed by (Lin et al., 2017). We used an embedding layer connected to a bidirectional Long Short-Term Memory (LSTM) with the self-attention mechanism followed by two fully-connected layers. The embedding layer is pre-trained with Skipgrams (Mikolov et al., 2013) extracted from the diabetics records, which have 5147 unique codes.

The main idea is to use the network to predict if a diabetic will have a complication on a time window, and also to use the self-attention mechanism to point most important records involved at the decision, so it can be used by medical doctors to understand the individual and population diabetes progress.

### 2.1 EXPERIMENTS

We made tests with 60, 120, 180, 240 and 360 days before the complication happen. The network input is restricted to the last twelve months, or less, of records, with a maximum limit of 500 and a minimum of 40 records. For each diabetic who have a complication we extracted one input sequence before the first record of complication, and for those who do not had complications we extracted one sequence, randomly selected, from each 24 months records. The low prevalence of complications makes the dataset imbalanced with only 133 positive cases. To overcome such problem, we over sample the training set on each test to achieve a perfect balance and avoid the major class overfitting.

For each time-gap we run a 5-fold cross-validation. To have a baseline we compared to a LSTM network without self-attention and a five layer Multi Layer Perceptron (MLP), both also with pre-trained embeddings. The LSTM have a hidden state of 64 and each MLP layer have 128 neurons. Table 1 report our results with the following metrics: area under ROC curve (AUC), Sensibility (Sen) and Specificity (Spc) based on thresholds estimated with Youden's index

Table 2: Top Attentions

| Average Attention | Frequency (%) + | - | Description |
|---|---|---|---|
| 0.01353 | 7.59 | 1.13 | Venous Color Doppler of Upper Limb |
| 0.01337 | 13.10 | 5.64 | Color Doppler of Aorta and Iliac |
| 0.01318 | 8.62 | 1.04 | Arterial Color Doppler of Upper Limb |
| 0.01290 | 29.66 | 10.72 | Arterial Color Doppler of Lower Limb |
| 0.01109 | 78.28 | 75.14 | Transthoracic Doppler Echocardiography |

The performance decrease as the time interval increased was expected, since the more critical the diabetic situation more obvious this becomes in the records due to the increase in specialized examinations and visits to the hospital\emergency. When comparing MedAttention with standard LSTM we can see its better ability to deal with longer inputs owing to the attention mechanism.

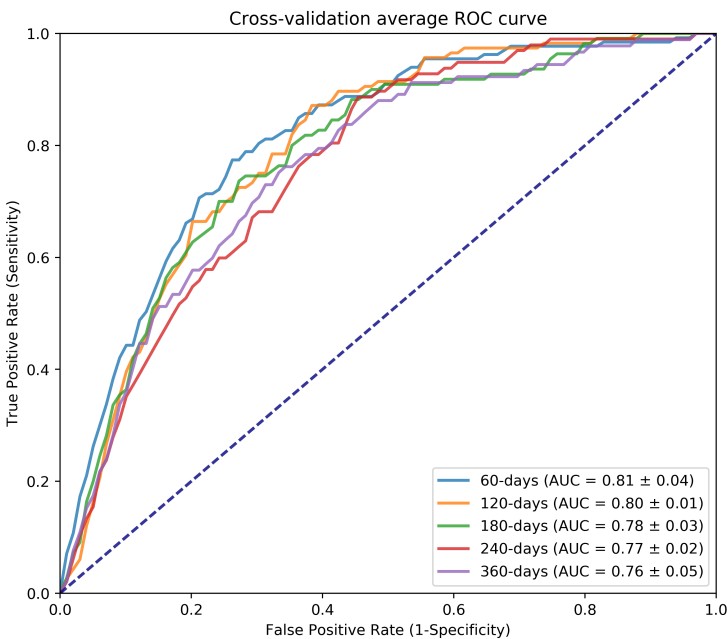

Figure 2: Mean ROC curve over cross-validation tests with MedAttention

The average ROC curves over folds at Figure 2.1 report the model good balance between positive and negative cases and also point a limitation of how hard is to set a threshold to ensure a low false positive rate.

Across all tests made we select some records with higher attention average as an attempt to understand what is most import to the network. Table 2 show us some of the top records with its frequency of occurrence over diabetics with (+) complications before it happened and without (-) complications. The high number of Doppler exams at diabetics with complications can be an evidence of his conditions.

## 3 CONCLUSIONS

We propose the use of self-attentive recurrent network to predict diabetes complications over financial medical data. Our results demonstrate how the architecture can be used to solve the problem, and also can be used to produce insights about the population analyzing the attention mechanism. The visualization and understanding of attentions is still under investigation and we expect to reach a way to report some analysis of cases with complications and the overall dataset.

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
