# OpenReview forum: "MedAttention: A Self-Attentive RNN to Predict Diabetes Complications with Financial Data"
_ICLR.cc/2018/Workshop — Reject_

### Official Review · AnonReviewer3 · 2018-03-10
**RNN + attention for prediction in diabetic cohort**

**Rating:** 5
**Confidence:** 5

**Review:**

The paper applies RNN+self-attention for prediction in the diabetic cohort. It shows attention helps in improving accuracy over LSTM and MLP. There are several concerns, however:
- Diabetes is a lifelong condition that takes decades to develop and result in complications. Limiting to just the past 12 months can be very limited. This could be due to the data limitation, but it should be addressed.
- The outcomes can be hard to determine and may or may not be directly caused by diabetes. Dialysis for example, is due to kidney failure, and it is not necessarily related to diabetes. Hence it is hard to say we can predict diabetes complications just by observational data.
- The RNN requires data sequencing. How was this done?
- The use of RNN + self-attention in healthcare has been tried recently: https://arxiv.org/abs/1707.05010
- The main usefulness of attention for healthcare is perhaps its interpretability, and this should be analysed. The authors have already recognized this limitation.

In conclusion, while this is a right direction to go for healthcare, much is needed to substantiate the results.

---

### Official Review · AnonReviewer1 · 2018-03-14
**Intriguing application of newer technique but below bar for quality**

**Rating:** 3
**Confidence:** 4

**Review:**

This is an interesting application of a recently proposed technique (structured self-attention [1]) to the problem of forecasting future complications in diabetic patient populations from medical claims data. The experimental results look promising, beating the next best baseline (an MLP) by an average of 10 points AUC across different forecasting windows ranging from 60 days to one year. However, the submission omits a many critical details about the experimental setting, and the baseline results are not persuasive. I think this is a valuable line of work, but it does meet ICLR's bar for quality, even for the workshop track. In the past, I might have advocated for acceptance based entirely on the merit of the application, but ICLR has seen similar work [2][3] in the last few years, and the novelty of the clinical time series has worn off.

Here is a laundry list of specific feedback, divided between strengths and weaknesses.

Strengths:

- Interesting application (forecasting future diabetes complications) with a large data set (300K patients, 56M time points)
- I like the claims data angle. The problems with claims are well-known, but the authors are right that they tend to be more ubiquitous and available: it is much easier to gather data sets of 100K-1M longitudinal patient records from claims than from dense EHR data.

Weaknesses:
- The data set is deceptively small: the actual cohort is NOT 300K patients but only 2,607 with diabetes, of whom only 133 have a complication! Can we really reliably train a complex neural net on such data? What are the actual sizes of the training, validation, and test splits? Do the authors use the full 300K data set for some kind of pretraining or transfer learning?
- The LSTM's results are suspiciously bad -- in most similar work [4][5] LSTMs clearly beat MLPs
- Is a plain LSTM the right baseline? What about an LSTM with plain attention, an attention-only model [6], or RETAIN [7]?
- How are the pretrained embeddings generated? The records (I assume, since this is not made clear) are sparse multi-hot vectors representing codes. Is some kind of pooling applied, or a technique similar to Dr. AI? How are the embeddings fed into the MLP?
- What about a baseline consisting of simple time-aware features plus an MLP?
- Given the small sample size of the data, it's critical to estimate the uncertainty of the results using, e.g., k-folds cross validation or bootstrapping.
- I'd recommend using the word "visit" or "encounter" (rather than "record") to refer to an individual time point -- less ambiguous.

[1] Lin, et al. A Structured Self-Attentive Sentence Embedding. ICLR 2017.
[2] Lipton, Kale, Elkan, and Wetzel. Learning to Diagnose with LSTM Recurrent Neural Networks. ICLR 2016.
[3] Bajor and Lasko. Predicting Medications from Diagnostic Codes with Recurrent Neural Networks. ICLR 2017.
[4] Choi, et al. Doctor AI: Predicting Clinical Events via Recurrent Neural Networks. MLHC 2016.
[5] Razavian, Marcus, and Sontag. Multi-task Prediction of Disease Onsets from Longitudinal Lab Tests. MLHC 2016.
[6] Vaswani, et al. Attention Is All You Need. NIPS 2016.
[7] Choi, et al. RETAIN: An Interpretable Predictive Model for Healthcare using Reverse Time Attention Mechanism. NIPS 2016.

---

### Decision · Program_Chairs · 2018-03-20
**ICLR 2018 Workshop Acceptance Decision**

**Decision:**

Reject

**Comment:**

Based on the reviews, this paper has not been accepted for presentation at the ICLR workshop. However, the conversation and updates can continue to appear here on OpenReview.